# Network-Based Analysis to Identify Drivers of Metastatic Prostate Cancer Using GoNetic

**DOI:** 10.3390/cancers13215291

**Published:** 2021-10-21

**Authors:** Louise de Schaetzen van Brienen, Giles Miclotte, Maarten Larmuseau, Jimmy Van den Eynden, Kathleen Marchal

**Affiliations:** 1Department of Plant Biotechnology and Bioinformatics, Faculty of Sciences, Ghent University, 9052 Ghent, Belgium; louise.deschaetzenvanbrienen@ugent.be (L.d.S.v.B.); giles.miclotte@ugent.be (G.M.); maarten.larmuseau@ugent.be (M.L.); 2Department of Information Technology, Faculty of Engineering and Architecture, Ghent University-IMEC, 9052 Ghent, Belgium; 3Department of Human Structure and Repair, Faculty of Medicine and Health Sciences, Ghent University, 9000 Ghent, Belgium; jimmy.vandeneynden@ugent.be

**Keywords:** network-based cancer data analysis, driver identification, metastatic prostate cancer, somatic mutations

## Abstract

**Simple Summary:**

The identification of cancer driver genes is, for statistical reasons, often biased toward genes that are altered frequently in a cohort. However, genes that are less frequently mutated can also alter cancer hallmarks. To detect such rarely mutated genes involved in driving metastatic prostate cancer, we analyzed the Hartwig Medical Foundation metastatic prostate cancer cohort. Hereto, we developed GoNetic, a novel network-based method that can detect genes with a lower mutational rate as members of recurrently mutated sets of genes connected on a prior interaction network. In contrast to state-of-the-art network-based driver identification methods, GoNetic retains information on sample-specific mutations and uses more properties of the prior interaction network. When applied to the Hartwig Medical Foundation cohort, GoNetic successfully prioritized both known drivers and rarely mutated driver candidates of metastatic prostate cancer. Comprehensive validation with other public data sets further supported the driver potential of these novel candidates.

**Abstract:**

Most known driver genes of metastatic prostate cancer are frequently mutated. To dig into the long tail of rarely mutated drivers, we performed network-based driver identification on the Hartwig Medical Foundation metastatic prostate cancer data set (HMF cohort). Hereto, we developed GoNetic, a method based on probabilistic pathfinding, to identify recurrently mutated subnetworks. In contrast to most state-of-the-art network-based methods, GoNetic can leverage sample-specific mutational information and the weights of the underlying prior network. When applied to the HMF cohort, GoNetic successfully recovered known primary and metastatic drivers of prostate cancer that are frequently mutated in the HMF cohort (*TP53*, *RB1,* and *CTNNB1*). In addition, the identified subnetworks contain frequently mutated genes, reflect processes related to metastatic prostate cancer, and contain rarely mutated driver candidates. To further validate these rarely mutated genes, we assessed whether the identified genes were more mutated in metastatic than in primary samples using an independent cohort. Then we evaluated their association with tumor evolution and with the lymph node status of the patients. This resulted in forwarding several novel putative driver genes for metastatic prostate cancer, some of which might be prognostic for disease evolution.

## 1. Introduction

Cancer driver identification depends on the genomic analysis of large cohorts of clinically well-characterized tumor samples. While gene-centric methods are ideal for identifying frequently mutated drivers in a cancer cohort, they often lack the power to identify more rarely mutated genes. Several studies showed that driver genes follow a long tail distribution, where their mutation rate can be lower than 3% [1,2,3]. To allow digging in the long tail of rarely mutated genes, network- or pathway-based methods search for recurrently mutated pathways or gene sets. These methods indirectly prioritize rare drivers by making use of the connectivity they display with more frequently mutated drivers [2]. Network-based methods use a network model to identify recurrently mutated subnetworks as proxies of recurrently mutated pathways. This network model is derived from a network prior in which the nodes are genes, and the edges represent interactions between genes [4].

When seeking to identify recurrently mutated subnetworks, several existing network-based methods propagate a gene-centric signal over the network prior. In this approach, gene-specific mutational information is mapped to nodes in the network prior by assigning a single score to each gene. This score is typically derived from a gene-centric driver identification method (e.g., ActiveDriver scores [5] or frequency scores) that is propagated over one single network (e.g., HotNet2 [3]). However, not all mutations in a given gene have the same driver potential. The use of a single score for each gene thus results in the loss of sample-specific information (i.e., HotNet2 [3], Hierarchical HotNet [6], NBS [7], MUNDIS [8]). Several tools exist to determine the mutation deleteriousness (e.g., CADD scores [9], FATHMM scores [10]) and provide sample-specific gene scores. In addition, only a few network-based driver identification methods can use the information embedded in the network model, such as the edge weights or directionalities [4].

To cope with these issues, we developed GoNetic, a novel network-based method that relies on probabilistic pathfinding. GoNetic takes as input a weighted interaction network and, for all samples in the cohort, a list of somatic mutations and their functional impact scores. Genes that are mutated in at least one sample are mapped on the interaction network, and the paths (i.e., series of consecutive edges in the network) between genes mutated in different samples are identified. Using this set of paths between somatic mutations, a subnetwork that connects as many pairs of relevant somatic mutations as possible is inferred. GoNetic is designed to efficiently handle data sets of up to thousands of cancer patients. Benchmarking GoNetic with other driver identification methods shows its performance and robustness are on par with what can be expected from the state-of-the-art.

Because drivers of metastatic prostate cancer are known to follow a long tail distribution [1], we applied GoNetic to identify novel drivers with a lower mutational rate of metastatic prostate cancer (mPCa). Metastasis is the leading cause of mortality among men with prostate cancer (PCa) [11]. Despite an initial benefit with androgen deprivation therapy, newer hormonal treatments, or cytotoxic chemotherapies, patients with metastatic disease eventually develop drug resistance and progress to lethal castration-resistant disease [11]. Prevention of metastatic disease depends on a more comprehensive identification of early drivers of mPCa. To discover novel mPCa drivers, we analyzed the HMF cohort, one of the largest cohorts on metastatic cancer [12]. GoNetic identified frequently mutated subnetworks that contain well-known primary and metastatic drivers as well as several rarely mutated driver candidates.

By exploiting the specific properties of complementary data sets, we could provide further support for the role of some of these more rarely mutated genes in driving mPCa. Some of the genes that were predicted as metastatic drivers by GoNetic were indeed found to be more frequently mutated in metastatic than in primary samples. Using matching primary and metastatic samples [13,14] allowed associating some of the predicted drivers with the early onset of metastatic disease. Using a cohort of primary samples with annotated lymph node status [1,15] allowed the identification of predicted drivers associated with a positive lymph node status. This further confirms their role in metastatic disease.

## 2. Materials and Methods

### 2.1. Discovery and Validation Cohorts

The discovery cohort is obtained from the HMF and consists of 352 tumor samples from patients with mPCa [12]. This cohort is further referred to as the HMF cohort. Samples were selected based on the following criteria: the primary tumor originated in the prostate, the sex of the patient is male, and the biopsy site is not from the primary tumor. This discovery cohort was used as a case study for GoNetic, after which we validated our findings using five independent publicly available data sets.

To determine which drivers prioritized by GoNetic were more significantly mutated in metastatic than in primary samples, we used data obtained from primary tumor samples from Armenia et al. [1]. We selected samples from patients that had a low risk of metastasizing at the time of the resection of their primary tumor. More specifically, we focused on the patients that: (1) did not show a new tumor event, (2) were free from tumor material after resection, and (3) were annotated as lymph node-negative according to the clinical information downloaded from TCGA. As such, 287 samples of the Armenia data set were selected, hereafter referred to as the TCGA cohort. We also used somatic mutation data from metastatic samples from Armenia et al. [1] as an independent metastatic cohort, hereafter referred to as the ARM cohort. This cohort originally consisted of 317 mPCa samples, of which nine with outlying mutation rates were removed.

To associate the potential driver genes with metastatic evolution, we used the data sets of Gundem et al. [13] and Kumar et al. [14]. Samples from the Gundem data set are whole-genome sequences (WGS), while those from the Kumar data set are whole-exome sequences (WES). Those data sets both contain multiple metastatic samples from the same patient for 26 patients. Additionally, for 16 patients, they also contained the matched-primary sample. Patients with only one metastatic sample were not considered in our analysis.

Finally, we tested whether some of the putative driver genes reported by GoNetic could be associated with positive lymph node (LN+) status. Hereto, we used the 69 primary samples from TCGA and 6 samples of the MPC project [15] with positive lymph node status. This cohort is further referred to as the LN+ cohort and compared to the TCGA cohort.

### 2.2. Novel Network-Based Driver Identification Method: GoNetic

GoNetic is a network-based driver identification tool, conceptually based on our seminal work [16,17,18] but redesigned to efficiently handle large cancer data sets. In general, the algorithm searches for short paths between a set *Y* of genes of interest, in our case: between genes containing somatic variants in different samples. It then heuristically selects a subnetwork *S = (V, E)* that maximizes the scoring function:*ΣP*(*path*(*g, Y*)*|S*) − *p|E|*(1)

Here, the sum is over all genes *g* in *Y*, and *P(path (g, Y)|S)* is the probability that a non-empty path between gene *g* and the other genes in the set *Y* exists in *S*, i.e., the existence of paths in the network contributes positively to the score. The term *p|E|* is an edge penalty *p* for each of the *|E|* edges in *S*. The penalty enforces the subnetwork to be parsimonious in the number of edges. This constraint leads to the preferential selection of paths that contain overlapping edges, and hence the algorithm finds frequently mutated subnetworks.

To determine the probability that a path exists between two nodes, edge probabilities are required. If the input network already includes edge probabilities, then these are used as initial weights; otherwise, all edges are initialized with a weight of 1. The edges are then reweighted by GoNetic based on the topology of the network model. From the distribution of out-degrees of the nodes in the interaction network, a power-law distribution is estimated. Then, all edges with start node *i* are reweighted with a sigmoidal factor:*s*(*i*) *=* (*1 + exp*(*3 ** (*out_degree*(*i*)*/cd**f_90 − 1*)))(2)
where *cdf_90* is the out-degree corresponding with the 90th percentile of the power-law distribution. The new edge weights are then recomputed as:*w*(*i, j*) *= 1* − (*1* − *1/s(i*)) *** (*1* − *w*(*i, j*))(3)

In this manner, edges connecting to hubs are downweighted, but they can still contribute to the final resulting network.

The score of a path is the product of the weights of its edges and can hence again be interpreted as a probability. These path probabilities are then further downweighted with sample-specific information derived from the functional impact score, the VAF, and a correction factor for samples with significantly higher mutation rates (mutation rate Z-score > 3.5). For a path from mutation *i* to mutation *j*, the path score is multiplied by the relevance scores and correction factors of the terminal nodes *i* and *j*.

GoNetic maps all mutations from the input data on the network and looks for paths between mutations in different samples. As such, for each mutated gene in the data set, a branch-and-bound algorithm constructs a path tree incrementally. In this path tree, the root corresponds with the mutated gene, and each path from root to leaf corresponds with a path in the network. For each node in the path tree, two scores are retained: the score of the current path and the maximal score that can be obtained by extending this path further, taking into account the weights of the next edges and the sample-specific weight factors of potential terminal nodes. Using these scores, a priority queue determines the next leaf from which the path tree should be extended.

To keep the computational complexity of the subnetwork selection in check, not all paths are retained. As such, the following filters are applied to the pathfinding procedure: only the paths with the highest scores are retained (default: max 25 paths), all paths with a path probability below a given threshold value (default: 0.2) are rejected. Additionally, the path length is limited to 4 edges. To avoid connecting mutations that are far removed from the network, longer paths are less likely to be biologically relevant [19,20]. The priority queue guarantees that high scoring paths are found early on, and as such, these filters are integrated into the branch-and-bound algorithm, allowing the early termination of the pathfinding procedure.

The subnetwork selection is then modeled as a decision-theoretic problem in the same way as in PheNetic [16,17,18]: first, for each mutation, all paths originating from this mutation are encoded as a conjunctive normal form and then compiled to a deterministic decomposable negation normal form (d-DNNF) with c2d [21]. Using these d-DNNFs, the probability of the existence of at least one path originating from this gene in a given subnetwork can be efficiently computed. To allow this approach to scale to the large cancer data sets, the subnetwork selection algorithm was parallelized and extended with a score cache. This cache encodes for each d-DNNF the interactions in the subnetwork that are relevant to this d-DNNF in a bit-vector, which is used as the key in the score map. Multiple distinct subnetworks can have the same set of relevant interactions for a given d-DNNF; as such, this cache allows computed probabilities to be reused for the scoring of multiple subnetworks.

The subnetwork selection is performed for a range of edge penalties. For each edge penalty, multiple runs are performed, and the resulting networks are compared based on score, resulting in network stability and size metrics. Edge penalties that result in either: (1) a negative score, (2) an average Jaccard index between the subnetworks smaller than 0.5, or (3) subnetworks containing more than 80 interactions are rejected. The union of the highest-scoring subnetworks of all the remaining edge penalties then forms the final resulting network of GoNetic. The genes in this network that are also mutated in the input data are compiled in a gene list, and the genes in this list are assigned a rank based on the highest edge penalty for which they appear in the resulting subnetworks. Genes that appear in a subnetwork with a higher edge penalty require more or stronger connections with other mutated genes through paths in the network, else the score of that subnetwork would be too low. As such, this approach prioritizes genes that occur in highly connected subnetworks while ranking more weakly connected genes lower.

We can distinguish two groups of genes in this resulting subnetwork: (1) mutated genes, i.e., the start and end points of the paths of interest in this subnetwork; and (2) connector genes, i.e., genes that are not themselves in the mutation data, but that are required to connect the mutated genes on the network.

### 2.3. GoNetic Benchmarking

We compared GoNetic with other driver identification methods using a well-established evaluation framework based on the study of Tokheim et al. [22]. GoNetic was compared with 2020+ (https://github.com/KarchinLab/2020plus, accessed on 25 May 2021), TUSON [23], OncodriveFML [24], MutsigCV [25], OncodriveClust [26], MuSiC [27], ActiveDriver [5], and OncodriveFM [28] using the benchmark result described in Tokheim et al. [22] study.

On the Tokheim data set, GoNetic was run with the default parameter settings, and Reactome (version 2018) [29], consisting of 207,668 interactions among 10,123 genes, was used as a network model. The mutations in the Tokheim et al. [22] benchmark data set were assigned functional scores using the combined annotation-dependent depletion (CADD [9,30]). CADD scores were used because they were already available for the genome build used in Tokheim et al. [22] study. To convert the CADD scores to probabilities, representing the impact of the mutation on the path scoring, a sigmoid function was used:*p =* (*1 + 1.04*^(*−x+10*)^)*^−1^*(4)
where *x* is the CADD score of a somatic mutation. To reduce the computational cost, only the mutations with a score over 70% were used to run GoNetic on the benchmark, resulting in a total of 58,567 mutations used as input.

Next to GoNetic, Hierarchical HotNet, another network-based driver identification method, was added to the benchmark results as part of the current study [6]. The Hierarchical HotNet results are poorer than expected, see Table 1. However, as this method is also a network-based approach that does not assign a gene-based *p*-value, it is not entirely clear how a threshold on significance should be set. The results of GoNetic and Hierarchical HotNet included in the benchmark analysis can be obtained in Appendix A.

To compare results, the Tokheim benchmark assumes that the output of the driver identification method is a gene list, with *p*-values assigned to each gene. Network-based driver identification methods typically identify modules or subnetworks of putative drivers rather than individual drivers, resulting in several driver genes receiving the same rank or *p*-value. To make GoNetic fit the Tokheim framework, we extract from the output network the nodes that had at least one mutation in the input data set and ranked them based on the highest edge penalty for which they were included in the final network, where a higher edge penalty corresponds to a lower *p*-value in other methods.

For each method, the performance was assessed by calculating the true positive ratio. This corresponds to the number of true positive predictions on the total number of predictions, where true positive predictions correspond to those driver genes that were already reported in the Cancer Gene Census (CGC). We used this metric rather than the AUC because predictions that are not present in CGC are not necessarily false positives. Indeed, metrics, such as AUC, that consider all the non-CGC predictions as false positives are biased toward methods that rely on prior information (e.g., supervised methods or methods that use a network prior). To calculate the true positive ratio for the methods that report a *q*-value, we set a threshold of 0.1 to determine the significantly selected genes [22]. For GoNetic and Hierarchical HotNet, the entire resulting mutated gene lists were used since these methods do not compute a gene-level *q*-value. To have a more comparable number of genes between the methods, we redid the analysis using for all methods the same number of genes, corresponding to the number of genes that received the highest rank with GoNetic (87 genes).

### 2.4. OnCompare: Procedure to Compare the Mutation Rate of Driver Genes between Sample Groups

To compute the statistical significance of the degree to which genes were more mutated in metastatic than in primary samples, we developed OnCompare. This statistical procedure copes with confounding factors such as a tumor mutational burden (TMB) and the tumor purity of a sample. This is important because these factors can cause differences in the number of mutations between groups of samples.

In general, OnCompare is a statistical testing procedure that can be used to compare the number of mutations in a gene, or in sets of genes, between two different groups of samples. Contrarily to commonly used tests, such as Fisher’s exact test or the binomial test, it relies on a Poisson binomial distribution, which accommodates for the presence of sample-specific confounding factors. The testing procedure consists of two components: (i) the assignment of sample-specific probabilities that compensate for confounders and (ii) the statistical test. To assign the probabilities of a gene being mutated in a sample, we rely on logistic regression to model the relation between a confounder in a sample and the probability that the gene is mutated in that sample. Typically, there exists a proportionality between confounders and mutation rate that is naturally captured by linear models. For instance, samples with a higher TMB are assigned a higher probability of containing a mutation in a predefined gene. Once the logistic regression model is fit to the data, it is used to assign to each sample a background probability. The test operates under the hypothesis that the probability of a sample having a gene mutated under the null hypothesis is independent of the group it belongs to. Here, this means that the null hypothesis assumes no intrinsic difference in mutation rate for gene *i* between the different sample groups. Denoting the probability of a sample harboring a mutation in gene *i* by *P(S_i_)*, the group by *G,* and possible confounders by *ϴ*, we can write this as:*P*(*S_i_|G,**ϴ*) *= P*(*S_i_|**ϴ*)(5)
where *P(S_i_|**ϴ)* is modeled using logistic regression, fit to all samples from both groups. When no confounders *ϴ* are known or measured, we have under the null hypothesis:*P*(*S_i_*) *=* (*m_1_ + m_2_*)*/*(*n_1_ + n_2_*)(6)
where *n_i_* denotes the number of samples in group *i* and *m_i_* the number of samples with a mutation in group *i*. Given the sample-specific probabilities, the number of mutations in both groups, *M_1_* and *M_2_*, each follow a Poisson binomial distribution. Because the mutation status in one group is independent of the other group, we can write the joint probability mass function as:*P*(*M_1_, M_2_*) *= P*(*M_1_*) *× P*(*M_2_*).(7)

To obtain a right-sided probability or *p*-value, we sum over all states that result in an equal or higher ratio than the observed ratio, *m_1_*/ *m_2_*, of mutations in group 1 to group 2:(8)$$\sum_{\left(v_{1},\;v_{2}\right)\epsilon V}P\left(v_{1}\right)P\left(v_{2}\right),\;V=\left\{\left(v_{1},v_{2}\right)\;|\;v_{1}/v_{2}\geq m_{1}/m_{2}\right\}.$$

Conversely, a left-sided *p*-value can be obtained by summing over all states:(9)$$\sum_{\left(v_{1},\;v_{2}\right)\epsilon V}P\left(v_{1}\right)P\left(v_{2}\right),\;V=\left\{\left(v_{1},v_{2}\right)\;|\;v_{1}/v_{2}\leq m_{1}/m_{2}\right\}.$$

OnCompare can also be applied to compare the mutation rates in sets of genes. To do so, we fit a logistic model to each of the genes separately and then determine a single probability that a sample contains a mutation in at least one of the genes in the set. Let *U* be a set of genes *{g_1_, …, g_|U|_}* and denote the probability of a sample being mutated in *g_i_* by *P(S_i_|**ϴ)*, then the probability of a sample containing at least one mutation in *U* is given by:(10)$$P\left(U|ϴ\right)=\;1\;-\;\overset{\left|U\right|}{\underset{i=1}{\prod }}(1\;-\;P(S_{i}\;|\;ϴ)).\;$$The resulting *p*-value then expresses the probability of the observed ratio of the number of samples in both groups that have at least one mutation in the set of genes *U*, under the null hypothesis that the intrinsic mutation rate is equal in both groups.

Without confounders taken into account, we have demonstrated that the test shows a suitable correspondence to Fisher’s exact test when comparing the difference in mutation rate between, respectively, the primary and HMF and ARM metastatic cohort (Appendix A).

### 2.5. Analysis of the Discovery Cohort

As a case study, we applied GoNetic to identify novel drivers of metastatic prostate cancer (mPCa) on the HMF cohort. Prior to applying GoNetic on the HMF cohort, several pre-processing steps were performed. Somatic variants obtained with the processing and analysis pipeline of the HMF were downloaded from the official HMF website. Somatic variants that did not pass all criteria to be called somatic were removed. We removed somatic variants with coverage below five reads and variant allele frequency (VAF) below 0.1.

To annotate somatic variants, we used the dbNSFP variant effect predictor (VEP) plugin [31]. We used the ‘pick’ option to ensure we prioritize per variant the most relevant annotation only (see details in VEP documentation). Silent mutations, mutations in introns, intergenic, upstream, downstream, 3′UTR and 5′UTR regions were discarded from the analysis.

Functional impact scores of the variants used to define the relevance scores in GoNetic were derived from FATHMM-MKL, which assigns pathogenicity scores to single nucleotide variants [10], and FATHMM-INDEL, which predicts the functional effect of small insertions and deletions (indels) [32]. Both tools assign a value between 0 and 1 to somatic variants where values closer to 1 correspond to a stronger functional impact of the mutation on the gene. Somatic variants that did not receive a FATHMM score were discarded from our analysis.

To identify samples with a deviating number of mutations, the distribution of the number of mutations per sample was log-transformed to obtain a normal-like distribution. Outlier samples were subsequently defined as the samples that fall below the boxplot’s lower whisker (Q1 − 1.5 × IQR) or above the boxplot’s upper whisker (Q3 + 1.5 × IQR). For the HMF cohort, 325 samples remained after filtering outliers.

Appendix A summarizes the number, the type, and annotations of somatic variants retained after filtering outliers. In total, we retained 325 samples from the HMF cohort containing 18,318 somatic mutations, of which 16,261 SNP and 2057 indels (596 insertions and 1461 deletions).

GoNetic was run on a human-specific interaction network, Reactome [29], and 18,318 somatic mutations of the HMF cohort. The frequency of the somatic variants was corrected for purity and ploidy by HMF, and the functional impact scores were determined with FATHMM predictions.

### 2.6. SomInaClust Analysis on the HMF Data Set

To identify mutational hotspots in the genes prioritized with GoNetic, we used SomInaClust [33]. As in the original publication, we ran SomInaClust with COSMIC as reference (v92) to obtain prior information on gene hotspots and background mutation rates. During the reference step, 7378 mutation hotspots were identified in 429 genes from the CGC list (v92) [34]. When searching for mutational patterns with SomInaClust, only frameshift and in-frame deletions or insertions, missense, nonsense, splice site, and silent mutations were considered [33]. SomInaclust determines, based on the cohort’s mutation data, gene-specific oncogene (OG) and tumor suppressor gene (TSG) random mutation probabilities (pOG and pTSG) [33]. A multiple testing correction of the pOG and pTSG is performed using the Benjamini–Hochberg method. The driver gene probability is calculated based on the product of the corrected *p*-values (qOG and qTSG), and genes with qDG ≤ 0.05 are defined as putative driver genes. In addition, to classify the putative driver genes as putative OGs or TSGs, the OG and TSG scores are defined. The OG score is the proportion of clustered OG mutations to the total number of OG mutations. In contrast, the TSG score is the proportion of TSG mutations on the total number of mutations. Genes are classified by SomInaClust as OGs when their OG score is above 20% and as TSGs when their TSG score is above 20%. When both the OG and the TSG score are above 20%, the driver is classified as a TSG [35].

### 2.7. Gene Ontology Enrichment Analysis

Gene ontology (GO) enrichment analysis was performed using the analysis option of the STRING v11 platform [36], focusing on the GO biological process terms. GO enrichment analysis was performed on the genes belonging to each subnetwork identified by GoNetic. Results are reported in Section A.1. Biological processes enriched with a false discovery rate (FDR)-adjusted *p*-value below 0.05 were considered as significant.

### 2.8. Validation of the Other Cohorts

To determine which drivers prioritized by GoNetic were more significantly mutated in metastatic than in primary samples, the mutational enrichment between, respectively, the two metastatic cohorts (HMF and ARM) and the primary cohort (TCGA) could be assessed (Figure 1, panel 1). Mutational enrichment comparisons were performed using OnCompare.

To maximize the comparability between, respectively, the WGS samples from HMF and the WES samples from ARM and TCGA, we performed the analysis only on somatic mutations detected in the HMF cohort that mapped to regions that were subjected to sequencing in the Armenia et al. [1] samples. Two percent of the somatic variants in coding regions originally detected in the HMF cohort could not be mapped to regions covered by the study of Armenia et al. [1]. For only one gene identified by GoNetic (i.e., MUC19), the mutations detected in the HMF cohort fell outside the regions covered by the Armenia et al. [1] study.

Appendix A shows that the TMB largely differed between the primary and both metastatic cohorts, but the tumor purity did not. Therefore, only TMB was considered as a confounder in our analysis. TMB was evaluated prior to applying any filtering.

Appendix A presents, for each gene reported by GoNetic, the OnCompare mutational enrichment *p*-values obtained with and without correcting for TMB for both the HMF and ARM metastatic cohorts.

All genes identified by GoNetic that were significantly enriched in mutations in at least one metastatic cohort with and without correction for TMB were kept for further analysis. Mutational enrichment signal in metastatic samples is divided into three categories. First, the strong signal corresponding to genes significantly enriched in both HMF and ARM metastatic cohorts even after TMB correction. Second, the intermediate signal corresponds to genes enriched in mutations in both the HMF and ARM metastatic cohorts without TMB correction and only in one after TMB correction. Third, the weak signal corresponds to genes enriched in mutations with and without TMB correction in only one cohort. In addition, the most likely primary drivers (i.e., genes that were not more mutated in the metastatic than in the primary samples but that were frequently mutated across the cohorts) were also retained. A distinction between genes that were never enriched in mutations in both metastatic cohorts and genes that were enriched in mutations in metastatic cohorts only before TMB correction was made. Three genes (i.e., GLI3, GAK, and SCN9A) were significantly enriched in mutations in both the HMF and ARM metastatic cohorts, but only without TMB correction. Although those genes were not mutated in the primary cohort, their low mutational frequency in the metastatic cohorts did not allow finding significant mutational enrichment signals in either of the metastatic cohorts. Therefore, these genes were no longer considered in the subsequent analysis.

To assess whether mutations in the retained genes prioritized by GoNetic could be associated with the evolution from primary to metastatic disease, we analyzed the Gundem and Kumar data sets. We searched for mutations in genes prioritized by GoNetic that occurred in different samples from the same patient (Figure 1, panel 2).

Genes mutated in the primary (if available) and in all matching metastatic samples for the majority of the patients (>50% of the patients containing a mutation in the gene under consideration in all of their metastatic samples) were considered as involved in the early onset of the metastatic disease or primary drivers. Genes for which no mutation was detected in the matching primary if it was available, but that contained the same mutation in the majority (>50%) of the metastatic samples (then referred to as truncal mutation) in most patients (>50%) were considered transitional drivers, as they must originate at the transition between the primary to the metastatic stage. Genes that were never found in the matching primary samples if available and not found truncal in most of the patients (so the mutation occurred in less than 50% of the matching metastatic lesions) were considered late drivers.

To test whether some of the putative driver genes reported by GoNetic could be associated with positive lymph node (LN+) status (Figure 1, panel 3), we used OnCompare without correcting for differences in TMB because the average TMB was comparable between primary samples of patients with positive and negative lymph node (LN−) status (Appendix A).

## 3. Results

### 3.1. GoNetic: A Flexible Network-Based Driver Identification Method

We developed GoNetic, a network-based method based on probabilistic pathfinding, designed to handle large cancer cohort sizes using large network priors.

The method takes as input a set of somatic mutations and a network prior to driving its analysis (Figure 2, panel A). In such a network prior, nodes represent genes, and edges represent interactions between genes. Edge weights can reflect any desired network property, e.g., the degree of belief one assigns to the edges. In our case, we opted for a topology-based weighting in order to reduce the bias toward highly connected genes. Genes in the network are annotated with sample-specific mutational information obtained from the input cohort. Subsequently, paths between mutated genes in different samples are enumerated, where a path is defined as a list of consecutive edges in the interaction network. The source and target nodes of a path thus correspond with two different sample-specific mutations. Paths between sources and targets that are mutated in the same samples are excluded, hereby assuming that multiple mutations in the same pathway in a single tumor are less informative than mutations in that pathway occurring in independent samples.

Once all paths between sources and targets have been derived, an optimization step infers a subnetwork that contains paths connecting as many pairs of sample-specific mutations as possible. However, the paths connecting these mutations should consist of a minimal number of edges. By imposing this parsimony in the number of edges, the algorithm searches for paths that contain as many overlapping edges as possible and hence detects subnetworks that are consistently mutated in independent samples in the cohort. These subnetworks are considered proxies of driver pathways.

During pathfinding and optimization, each path is assigned a probability, which reflects the degree of belief that the path is associated with the carcinogenic phenotype under study. This probability takes into account the weights of the edges composing the path and properties that reflect the driver potential of the connected source and target mutations, such as sample-specific functional impact scores and VAF (Figure 2, panel B). To avoid that samples with a higher mutation rate would skew the search for paths in the pathfinding step toward these populations with higher mutation rates, the scores are corrected based on the mutation rate of the sample in which they occurred. Because the probabilities of the paths are accounted for, GoNetic does not only prioritize relevant genes but also extracts the most relevant edges. In this way, GoNetic allows extracting the most important network context in which the potential driver mutations occur.

### 3.2. GoNetic Benchmark with the State-of-the-Art

To compare the performance of our method with that of some of the well-known state-of-the-art driver identification methods, we performed a benchmark using the evaluation framework of Tokheim et al. [22] (Materials and Methods).

The performance of each method was assessed by the true positive ratio. This corresponds to the number of true positive predictions on the total number of predictions, where true positive predictions correspond to those driver genes that were already reported in the CGC list. The higher the true positive ratio, the more the predictions made by a method are enriched in known cancer genes. For each method, Table 1 shows the overlap between their identified drivers and the CGC list; a larger overlap suggests that the predictions made by a method are more enriched in known cancer genes. Table 1 (panel A) considers for all methods that report a *q*-value, the true positive ratio for the predictions obtained with a *q*-value < 0.1. The network-based methods GoNetic and Hierarchical HotNet prioritize subnetworks containing several genes at once and hence do not provide a gene-level *q*-value. For these methods, the entire prioritized gene lists were used.

However, as with a certain *q*-value, the number of prioritized genes differed largely between the different methods. Table 1 (panel B) shows the true positive ratio considering this time for all methods the same number of genes, corresponding to the number of genes that received the highest rank with GoNetic (87 genes). Because Hierarchical HotNet does not provide a gene ranking, it could not be included in this comparison. This restriction to the top 87 genes resulted in a larger overlap ratio with CGC, indicating that all tools, including GoNetic, correctly prioritize known drivers.

Table 1 and Figure 3 show that the obtained true positive ratio depends on the properties of the methods: supervised driver identification methods (i.e., 2020+, TUSON), which use next to sequence-derived also pathway-based information, show a relatively higher true positive ratio than unsupervised methods that use merely sequence-derived information (i.e., MuSiC, MutSigCV, OncoDriveFM, etc.).

Expectedly, GoNetic has a true positive ratio falling between the true positive ratio of the supervised and the unsupervised methods that rely on frequency only (e.g., MuSiC) as it is not biased by training data but still uses a prior network to drive its analysis. In general, the performance of network-based driver identification methods is expected to be underestimated on gold standards such as CGC since network methods prioritize subnetworks containing several genes at once. Some of these genes are frequently mutated and likely already described in CGC, but the infrequently mutated ones are mostly unknown. Unknowns are considered false positive in the benchmark, even though they might be true drivers. However, Figure 3 also shows that most methods, including GoNetic, agree at least to some extent on the strong signals in the data. Indeed, mutual comparison between the predictions made by each of the different methods shows that the overlapping predictions (Figure 3, upper triangle) mostly correspond to the predictions that also overlap with CGC genes (Figure 3, lower triangle). Methods that are more similar in their underlying methods and in the information they use (e.g., TUSON and 2020+) tend to overlap more. For instance, GoNetic mostly overlaps with OncodriveFM, OncodriveFML, and the supervised methods (i.e., TUSON and 2020+). These methods and GoNetic all use the frequency of the mutations and their functional impact scores.

### 3.3. Application of GoNetic to the HMF Data Set

To discover mPCa drivers, we analyzed the HMF cohort, one of the largest cohorts on metastatic cancer. GoNetic identified 14 recurrently mutated subnetworks, consisting of 75 interactions and 87 genes, of which five were connector genes (genes that were not mutated themselves). The vast majority of mPCa samples from the HMF cohort contained at least one mutation in any of those 82 putative driver genes (306/325, 94.15%). GoNetic assigns a rank to its identified drivers based on their contribution to the subnetwork selection (see Materials and Methods). Several of the top prioritized genes (i.e., *TP53*, *AR*, *SYNE1*, *MUC16*, *FOXA1*, *APC*, *RYR2*, *KMT2D*, *KMT2C*, *RB1*, *BRCA2,* and *SPOP*) correspond to the more frequently mutated genes in the HMF cohort (see genes in green in Appendix A and in blue in Appendix A). Moreover, the large overlap detected between the top-ranked driver genes prioritized by GoNetic and the ones identified by previous studies on the HMF cohort further confirms that GoNetic is able to capture the most clear signals in the data (Section A.2).

In addition to identifying the most significant signals in the data, GoNetic can dig into the long tail of rare drivers: 18 of its prioritized drivers were part of recurrently mutated subnetworks driven by one of the more frequently mutated drivers but were infrequently mutated themselves (mutated in less than 1% of the samples, highlighted in red in Appendix A). Subnetworks identified by GoNetic were ranked based on the number of samples with at least one mutation in the subnetwork. Somatic mutations in subnetworks 1 and 2 cover a large number of samples in the HMF cohort, respectively 76% for subnetwork 1 and 30% for subnetwork 2. These two subnetworks also appeared to be centered around frequently mutated and well-known drivers of mPCa, respectively *TP53* and *AR* [1,12,14,37,38] (see left panel of Figure 4 for the mutational plot of subnetwork 1, for the other subnetworks, see Appendix A panels A to N).

Most of the samples have mutations in multiple subnetworks (207/306 samples with mutations in genes prioritized by GoNetic, 67.65%, (Figure 4, right panel)), suggesting that hitting multiple subnetworks is a prerequisite to acquiring carcinogenic properties.

We also assessed whether genes within the same subnetwork were mutated in a specific metastatic tumor site (i.e., lung, liver, or bone). Indeed, this would indicate that the pathway reflected by the subnetwork is acting as a driver pathway in different sites but is affected in each site through a different gene or set of genes (Appendix A). Overall, 18 genes were at least once reported as significantly differentially mutated between different metastatic sites (Appendix A).

Except for three subnetworks (subnetworks 9, 11, and 14, which each contained two genes only), all other subnetworks were found to be enriched in several GO biological processes, including processes playing a crucial role in either cancer progression or metastasis (Appendix A and Appendix A, panels A to N).

From the 82 prioritized genes, SomInaClust identified mutational hotspots in only 12 of them (Section A.3, Appendix A). The fact that only a restricted number of genes prioritized by GoNetic were identified to contain mutational hotspots is expected, given their low mutational frequency in the HMF cohort.

To further validate the role of these prioritized genes in mPCa, we designed a strategy based on the analysis of complementary cohort data.

### 3.4. Distinguishing Primary from Metastatic Drivers Using Mutational Enrichment Analysis

First, we assessed the degree to which the identified genes were more mutated in metastatic than in primary samples. This allowed distinguishing the primary from the metastatic drivers. Because no matching primary samples for the metastatic tumors were available in HMF, we used primary samples reported in the study of Armenia et al. [1] originating from TCGA (Figure 1, panel 1). We hereby focused on primary samples that were the least likely to already contain metastatic drivers (see Materials and Methods). We performed the same enrichment analysis using an independent metastatic cohort described in Armenia et al. [1] and compared it with the same TCGA samples as a primary reference set. When assessing the difference in mutation frequency of the driver genes between the primary and metastatic cohorts, differences in tumor mutational burden (TMB) was accounted for (see Materials and Methods). The use of TMB as a cofactor affects the *p*-values (Appendix A), which is to be expected given the much higher TMB in metastatic than in primary samples.

According to the mutational enrichment analysis, the genes *TP53*, *AR*, *RB1*, *CTNNB1*, *CSPG4,* and *APC* exhibit a strong mutational enrichment signal in metastatic samples, both in the HMF and in the ARM metastatic cohorts, even after correcting for TMB. Except for *CSPG4*, these genes were also the most frequently mutated genes in the cohort and contain the well-known drivers of mPCa (i.e., *TP53*, *AR*, *CTNNB1,* and *RB1*) [1,12,14,37,38] (Table 2, 1st column).

For the remainder of the drivers prioritized by GoNetic, the lower mutational frequency limits the significance level for mutational enrichment that can be reached with the current sizes of the ARM and HMF cohorts, especially when also correcting for TMB. Table 2 (columns) summarizes how driver genes were categorized according to their level of significant enrichment between metastatic and primary samples.

Genes that were significantly more mutated in metastatic than in primary samples in both metastatic cohorts without TMB correction and in one metastatic cohort after TMB correction were labeled as genes with an intermediate mutational enrichment signal in metastatic samples. To this category belong, next to the genes that are mutated in at least 5% of the cohort (*BRCA2*, *MUC16*, *FAT4,* and *DCHS2*), several more rarely mutated genes (*MUC4*, *VCAN*, *TAF1L*, *CACNA1H*, *IGFR2*, *MUC2,* and *SHANK1*). Interestingly, *BRCA2*, a previously described metastatic driver of PCa, belonged to this category. *BRCA2* affects the DNA repair system and hence is associated with samples that have a high TMB [39]. Hence, accounting for TMB as a confounding factor might dilute the signal too much.

Genes that were significantly more mutated in metastatic samples in one of the metastatic cohorts after correction for TMB but that could not be confirmed by the other metastatic cohort with or without correction for TMB were categorized as genes with a weaker mutational enrichment signal in metastatic samples. To this category belong also rarely mutated prioritized drivers *LAMA2*, *MYH11*, *AMPH,* and *LRRC7*.

At the other end of the spectrum, we have genes that were clearly not more mutated in the metastatic than in the primary samples. To this category belong the frequently mutated genes, i.e., mutated in more than 4% of the samples over all cohorts. Here, a statistical signal could have been detected, but no mutational enrichment signal was found, not even after omitting the correction for TMB. Therefore, these genes are most likely primary drivers. Indeed, the genes prioritized by GoNetic that fall in this category correspond to previously described primary drivers, i.e., *SPOP*, *FOXA1,* and *PTEN* [40,41].

The last category is genes that are relatively frequently mutated in both metastatic cohorts and for which an enrichment in mutations in metastatic samples can be detected in both the HMF and ARM metastatic cohorts, but only without correcting for TMB (*KMT2D* (22), *KMT2C* (20)). These genes are frequently mutated but not significantly enriched in mutations in metastatic samples. This suggests that they are likely primary or potentiating drivers of metastasis that are already frequently present in the primary tumor.

Finally, we applied OnCompare at the subnetwork level rather than at the gene level. Five subnetworks (1, 2, 5, 7, and 12) were significantly more mutated in metastatic samples than in primary samples in both the HMF and ARM metastatic cohorts (Appendix A). For each of these subnetworks, a lower *p*-value was obtained compared to *p*-values of any gene in the subnetwork. These results highlight that the genes in the subnetworks each affect a different set of metastatic samples and act as independent drivers. For subnetworks 1 and 2, this is to be expected since they contain several genes that are themselves significantly differentially mutated between the primary and the metastatic samples (*TP53*, *APC*, *RB1,* and *CSPG4* for subnetwork 1, and *AR* and *CTNNB1* for subnetwork 2). In addition, subnetworks 5 and 12 illustrate that rarely mutated genes that lack statistical power at the gene level can still be significant at the subnetwork level.

### 3.5. Association of the Identified Drivers with Tumor Evolution

Secondly, we used the data of Gundem et al. [13] and Kumar et al. [14] to assess whether the drivers identified by GoNetic could be associated with the evolution from primary toward metastatic disease. Both studies provide data for multiple metastatic samples in the same patients and sometimes also for the matching primary samples. This setting allows distinguishing drivers involved in the early stages of metastatic disease from those relevant in later disease stages.

For only a few genes prioritized by GoNetic, mutations detected in a metastatic sample could also be recovered in the matching primary sample. This can be explained by the small size of the set of patients with matching primary samples (16 patients). In addition, none of the patients carried in their primary lesion a mutation in the primary-specific drivers identified by GoNetic (*SPOP*, *FOXA1*, and *ATM*, Table 2). Moreover, due to heterogeneity of the primary tumor, the clone that seeded metastasis might have been absent in the primary lesion that was selected for sequencing. Because only a few mutations were recovered in the primary samples, we also considered whether mutations were found to be truncal to metastatic lesions of the same patient. Indeed, these can serve as an additional indication of their involvement in primary disease or of potentiating later metastatic disease.

So, all genes for which mutation data were available were categorized in one of the following three classes: primary or early metastatic drivers, transitional metastatic drivers, and late metastatic drivers. This classification is based on the occurrence of their mutations in the primary and metastatic lesions (see Materials and Methods).

Genes belonging to these categories and those derived in Section 3.4 are summarized in Table 2.

Primary or early metastatic drivers (Table 2, rows) are driver candidates derived from the Gundem and Kumar data sets. These drivers are detected in the primary sample (if available) and truncal to the metastases in the majority of the patients. Expectedly, to this category belong the well-known primary drivers *FOXA1*, *PTEN* [40,41], and *ATM*. Interestingly, mutations in *KMT2D* and *KMT2C* were detected in primary samples but not recovered in all metastatic samples of the same patient. This suggests that during metastatic spread, mutations in those genes were not selected. The following genes with mutational overrepresentation in metastatic samples also qualified as early metastatic drivers: *TP53*, *RB1*, *CTNNB1,* and *DCHS2.* These genes might be suitable markers for disease progression, as they are likely metastatic drivers and they are already detectable in the primary tumor. In contrast, *MUC2* was not recovered in all matching metastatic samples, suggesting this gene might not be as important for metastatic spread.

Mutations in genes prioritized by GoNetic that were not detected in the primary sample, but found to be truncal to the metastases in the majority of the patients, were considered transitional metastatic events (i.e., *SYNE1*, *MUC16*, *TAF1L,* and *VCAN*). Because of their subclonal nature, they might remain undetected in the primary sample, or they might play a role in the transition toward metastasis. To this category belong the genes with an intermediate signal of metastatic overrepresentation.

A representative example of a late metastatic driver is *AR*, which is very rarely found truncal to matching metastatic tumors of the same patient and is never detected in the matching primary tumors. *LAMA2* shows a similar mutational profile as *AR:* it is enriched in mutations in metastatic samples but non-truncal. Mutations in *FAT4* and *CACNA1H* were found non-truncal in half of the patients. All these drivers are more likely involved in conferring adaptive phenotypes in later disease stages (such as disease resistance).

*APC* and *BRCA2* are never observed in any of the Gundem or Kumar metastatic samples, despite their rather high mutational frequency in both the HMF and ARM metastatic cohorts. Similarly, the potential metastatic drivers *MUC4*, *SHANK1*, *CSPG4*, *IGF2R*, *PCDHA8*, *AMPH*, *LRRC7,* and *MYH11* were not mutated in any of the Gundem or Kumar samples. These genes are harder to detect in the validation data sets, because of their small mutation rate and because of the small size of the validation data sets.

### 3.6. Identifying Drivers Prognostic of Disease Staging

Lastly, we assessed whether the mutations in the genes identified by GoNetic were associated with the lymph node status of the patient. Patients with lymph node metastases at the time of diagnosis (LN+ patients) usually have a worse prognosis than patients without lymph node metastases (LN− patients). An association with lymph node status, therefore, also indicates a potential role in driving metastatic disease. Table 2 shows with an asterisk the genes that were significantly more mutated in LN+ patients than in LN− patients. Appendix A shows the OnCompare *p*-values for all genes identified by GoNetic. Despite the relatively small size of the lymph node-positive data set and the highly unbalanced number of samples (75 LN+ versus 284 LN− samples), the observed enrichment of mutations in LN+ patients was significant for several genes (i.e., *TP53*, *TAF1L*, *BRCA2,* and *MYH11*, *p*-value < 0.05). Even though the size of the data set lacks power, some of these genes, i.e., *TAF1L* and *MYH11* (*p*-value < 0.05) and *SHANK1*, *MUC2*, *IGF2R,* and *CSPG4* (*p*-value < 0.1), were never observed mutated in the TCGA cohort (LN−). In contrast, known and likely primary drivers (i.e., *SPOP*, *FOXA1*, *PTEN,* and *ATM*) show the least significant *p*-values. With the exception of *FAT4*, none of the late metastatic drivers were more mutated in primary tumors of LN+ than in primary tumors of LN− patients.

## 4. Discussion

In this work, we present GoNetic, a flexible network-based driver identification method based on probabilistic pathfinding. GoNetic does not only prioritize driver genes. It also selects the edges from the network prior that are most relevant under the conditions investigated. Hereto, it integrates the network structure, the edge weights, and the sample-specific node weights. GoNetic can handle large tumor cohorts and performs in line with the state-of-the-art.

Applying GoNetic on the HMF cohort showed how the method successfully recovered both known PCa drivers (*SPOP*, *FOXA1,* and *PTEN* [40,41]) and known mPCa drivers (*TP53*, *AR*, *RB1*, *APC,* and *CTNNB1* [1,12,14,37,38]). These genes carry a strong frequency-based driver signal in the HMF cohort. Most were also picked up using SomInaClust and reported by van Dessel et al. [12]. Besides agreeing with previous studies on the statistically strong signals in the data, GoNetic also identified more rarely mutated driver candidates (genes mutated in less than 5% of the samples, i.e., in fewer than 16 samples out of 325 samples). These genes were identified as members of 14 subnetworks, several of which reflect processes related to mPCa disease. The first subnetwork, centered around *TP53*, is enriched in the cell cycle, DNA damage response, and apoptosis. The second network, centered around *AR* and *CTNNB1,* is associated with, among others, the androgen receptor signaling pathway and cell-cell adhesion. Interestingly, the third subnetwork, which is enriched in calcium-dependent cell-cell adhesion processes, mainly consists of protocadherins and cadherins. Cadherin disruption is known to impact tumor progression, cancer cell invasion, and metastasis [42,43]. The fourth subnetwork, centered around *RYR2*, is related to the regulation of the membrane potential, regulation of calcium concentration, and calcium transmembrane transport. Dysregulation of calcium ion signaling and transport remodeling is known to promote cancer cell proliferation [44,45]. In addition, in PCa, the inhibition of *RYR2* has been shown to result in the release of calcium ions and the protection of the cells against apoptosis [46]. The fifth subnetwork is enriched in O-glycan processing and cell adhesion. Aberrant glycosylation is a cancer hallmark and influences cancer progression [47,48]. The sixth subnetwork is enriched in processes that relate to extracellular matrix (ECM) organization and cell adhesion. The seventh subnetwork is enriched in membrane organization and endocytosis, which has recently been reported as a potential regulator of tumor metastasis [49]. The other subnetworks were not enriched in any relevant processes.

SomInaClust mutational hotspot analysis could confirm one novel tumor suppressor not yet identified in GCG: *SYCP1*. Aberrations in *SYCP1* have been associated with progression toward metastatic disease in castrate-resistant PCa [50].

To further validate the role of these prioritized genes in mPCa, we performed three additional analyses. Firstly, we assessed the degree to which the identified genes were more mutated in metastatic than in primary samples. Secondly, we assessed whether they could be associated with tumor evolution. Finally, we evaluated whether genes could be associated with the lymph node status of the patients.

Based on these analyses, prioritized drivers were subdivided into primary versus metastatic drivers and classified according to their putative time of origin during tumor evolution. Using this scheme, known primary and metastatic drivers were classified as expected, validating the results of the categorization scheme. Indeed, known primary drivers did not show a signal of mutational enrichment in metastases, whereas known metastatic drivers did. Moreover, known primary drivers were found to originate early during tumor evolution, whereas the time of origin of known metastatic drivers was in line with what was reported in the literature, i.e., either being early metastatic (*TP53*, *RB1,* and *CTNNB1*) or late adaptive (*AR*). In addition, *TP53* was found to be significantly associated with a positive lymph node status (*p*-value < 10^−5^) and hence could be used as an early prognostic factor.

Given the low mutation rate of the rarely mutated genes prioritized by GoNetic, the size of most available validation cohorts limits the power of detecting significant validation signals. Despite this, using these meta-analyses, we found for most of the genes a further validation of their role in metastatic disease in at least one of the validation cohorts. For instance, *TAF1L* is identified as a strong metastatic driver originating during the transition from primary to metastatic disease. In addition, it was found to be significantly associated with a positive lymph node status and hence might be prognostic for advanced disease. Increased expression of the homolog of *TAF1L* (*TAF1*) has been associated with the progression of human PCa to the lethal castration-resistant state [51]. Given the close homology between *TAF1* and *TAF1L*, it was hypothesized that *TAF1L* may have similar regulatory functions in cancer [52].

Despite not being present in the Gundem or Kumar data sets, *CSPG4* was strongly overrepresented in metastatic samples in both the HMF and ARM metastatic cohorts. In addition, it was associated with the positive lymph node status of PCa patients (*p*-value < 0.1), even though this association is weak. Previous studies have shown that *CSPG4* plays an important role in tumor cell proliferation and migration, as well as with poor prognosis and relapse in breast cancers [53].

*MUC16* and *SYNE1* are relatively frequently mutated in the HMF cohort, showing a mutational enrichment signal even after TMB correction in the cohort. They are classified as transitional drivers and *SYNE1* associated with the patient’s positive lymph node status (*p*-value < 0.05). Mutations in both genes have previously been associated with PCa in young men, supporting their putative role in advanced disease [54].

For most of the rarely mutated genes, their role in metastatic disease was supported by only one of the meta-analyses. For instance, *MYH11* was not detected in the Gundem or Kumar data sets used for the association with tumor evolution. Furthermore, it was not detected as enriched in mutations in metastatic samples using the correction for TMB. In contrast, it was found to be significantly associated with the positive lymph node status of the patients. *MYH11* has a role in cell migration and interacts with cell adhesion proteins; additionally, mutations in *MYH11* have been associated with several cancer types [55].

Surprisingly, *APC* and *BRCA2* are not observed in any of the Gundem or Kumar metastatic samples, despite their rather high mutational overrepresentation in the HMF and ARM metastatic samples. This indicates that they might be representative of a subset of mPCa patients that are underrepresented in the Gundem and Kumar data sets but prevalent in the HMF and ARM metastatic cohorts.

## 5. Conclusions

In this study, we present GoNetic, a network-based analysis framework to perform cancer cohort analysis. GoNetic allows exploiting large cohorts of sample-specific mutational information and properties of the network prior to identifying driver candidates. Being a flexible framework, GoNetic is easily extendable to other data sets (e.g., to combine mutational information with and expression data).

Analysis of the HMF mPCa cohort illustrates the potential of GoNetic in identifying novel drivers. Further validation of those driver candidates through meta-analyses resulted in forwarding several novel putative driver genes for mPCa, some of which might be prognostic for disease evolution.

## Figures and Tables

**Figure 1 cancers-13-05291-f001:**
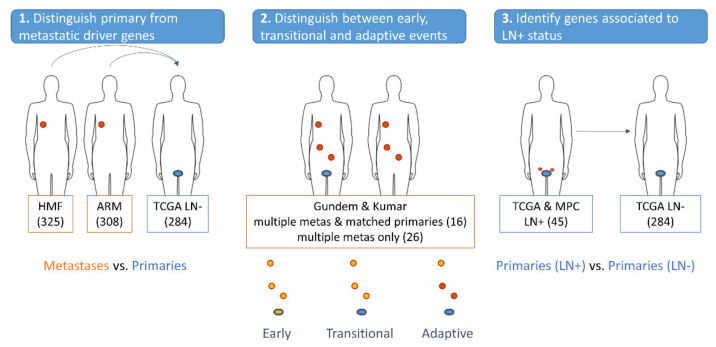
Validation of genes identified by GoNetic. **Panel 1**. First, we compared the mutation rate of genes prioritized by GoNetic between each of the metastatic cohorts (HMF and ARM) and the primary cohort (TCGA). Mutational enrichment signal in metastatic samples is divided into three categories: strong, intermediate, and weak. **Panel 2**. Secondly, we assessed whether GoNetic drivers could be associated with tumor evolution using cohorts of matching primary and metastatic samples. Genes reported by GoNetic were defined as primary/early drivers, transitional drivers, and late/adaptive drivers. **Panel 3**. Lastly, we assessed whether mutations in any of the genes prioritized by GoNetic could be associated with the lymph node-positive status of the patients.

**Figure 2 cancers-13-05291-f002:**
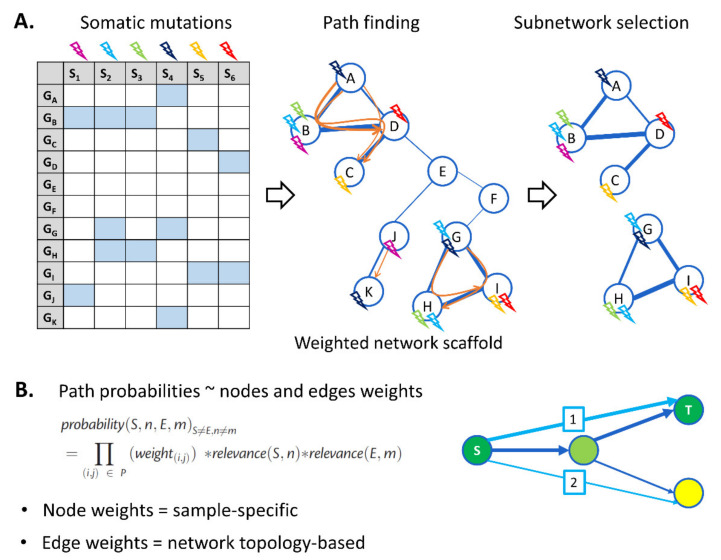
GoNetic workflow. **Panel A**. GoNetic uses as input somatic mutations of a cohort of tumor samples and a network prior. Somatic mutations occurring in each of the samples are mapped to the genes in the network (mutations occurring in the same sample are represented in the same color). During pathfinding, GoNetic identifies paths between genes that are mutated in different samples, displayed in orange. During the optimization step, GoNetic selects subnetworks that connect as many genes as possible with highly probable paths while using a minimal number of edges. **Panel B**. To each path, a probability is assigned that is a function of the node and edge weights. Nodes are weighted based on sample-specific information (i.e., the variant allele frequency (VAF), the functional impact score, and the sample mutation rate). Nodes with high weights are indicated in green, and those with lower weights in yellow. Edges are weighted according to the connectivity of the network prior. The line width reflects the edge weight. The example graph illustrates how path 1 will be assigned a higher probability than path 2 because it connects a source (S) to a higher weighted target (T) node and because the edges that make up the paths also have a higher weight than the one used to constitute path 2.

**Figure 3 cancers-13-05291-f003:**
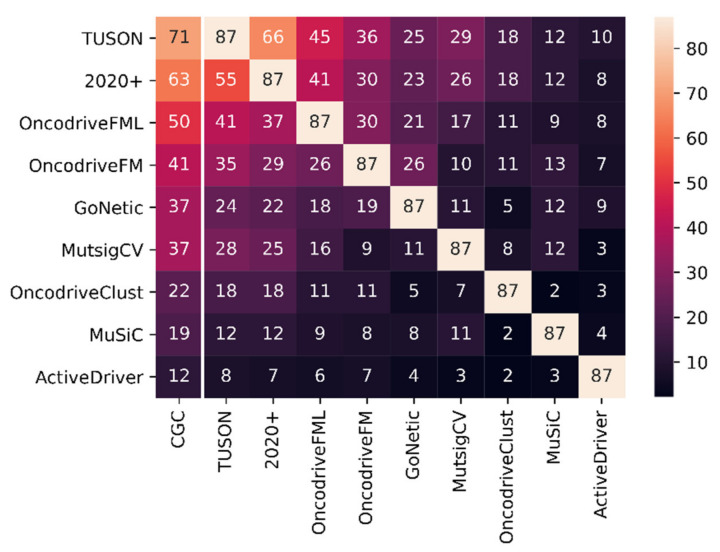
CGC benchmark heatmap between methods. Left-most column: intersection between the top 87 genes identified by each method and CGC. Upper triangle: intersection of the top 87 genes between methods. Lower triangle: intersection between the CGC genes in the top 87 genes between methods.

**Figure 4 cancers-13-05291-f004:**
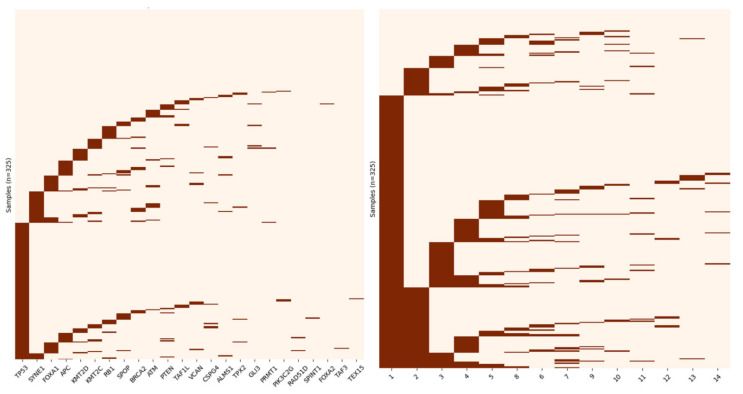
**Left panel**. Distribution of mutations in the HMF cohort in the first subnetwork identified by GoNetic. **Right panel**. Distribution of the samples (*Y*-axis) that carry mutations in any of the 14 subnetworks identified by GoNetic (*X*-axis). Samples that have at least one mutation in any of the 14 subnetworks are displayed.

**Table 1 cancers-13-05291-t001:** CGC benchmark of different driver identification methods. #CGC: number of identified drivers that are in the gold standard CGC, #significant: number of driver genes prioritized by each method on the Tokheim data set. % CGC: positive predictive value or the number of true positive predictions (overlap with CGC) on the total number of predictions. Panel A. Considering, for each method, all the significantly identified drivers. Panel B. Considering, for each method, the top 87 predicted genes.

	Panel A			Panel B		
	#CGC	# Significant	% CGC	# CGC	# Significant	% CGC
TUSON	115	243	0.4733	71	87	0.8161
2020+	104	208	0.5000	63	87	0.7241
OncodriveFML	107	679	0.1576	50	87	0.5747
OncodriveFM	143	2600	0.0550	41	87	0.4713
GoNetic	81	331	0.2447	37	87	0.4253
MutSigCV	71	158	0.4494	37	87	0.4253
OncodriveClust	59	586	0.1007	22	87	0.2529
MuSiC	173	1975	0.0876	19	87	0.2184
ActiveDriver	34	417	0.0815	12	87	0.1379
Hierarchical HotNet	41	455	0.0901			

**Table 2 cancers-13-05291-t002:** Classification of the driver genes identified by GoNetic according to the mutational enrichment in metastatic cohorts (columns) and their association with tumor evolution (rows). **Columns**: Mutational enrichment signal in metastatic samples is divided into five categories (gray background): strong = genes significantly enriched in mutations in the HMF and ARM cohorts even after TMB correction, intermediate = genes significantly enriched in mutations in both the HMF and ARM cohorts without TMB correction but in one cohort after TMB correction, weak = genes significantly enriched in mutations with TMB correction in one metastatic cohort only. The two last categories correspond to frequently mutated genes that i) were enriched in mutations in both metastatic cohorts but only without TMB correction (no sign. enrichment with TMB correction) and ii) for which no enrichment at all could be detected, even not without correcting for TMB (no sign. enrichment). **Rows**: The primary/early metastatic category corresponds to genes for which mutations were present in the primary tumor (when available) and truncal to the matching metastases in most of the patients. The **transitional** category corresponds to genes for which mutations were truncal in the majority of the patients but not present in the primary tumor. The late category corresponds to genes for which mutations were non-truncal to metastatic lesions and never detected in the matching primary tumor.

	Drivers with Mutational Overrepresentation in Metastatic Samples			Primary Drivers or Genes Potentiating Metastasis	
	Strong	Intermediate	Weak	No Sign. Enrichment with TMB Correction	No Sign. Enrichment
Primary/early metastatic drivers	*TP53* **	*DCHS2* *		*KMT2D*	*FOXA1*
	*RB1*	*MUC2* *		*KMT2C*	*SPOP*
	*CTNNB1*				*ATM* *PTEN*
Transitional metastatic drivers		*SYNE1* * *MUC16*			
		*TAF1L* ** *VCAN*			
Late metastatic drivers	*AR*	*FAT4*	*LAMA2*		
		*CACNA1H*			
Not present in Gundem nor in Kumar	*APC* *CSPG4* *	*BRCA2* ** *IGF2R* * *MUC4*	*MYH11* ** *LRRC7* *AMPH*		
		*SHANK1* *			

* = genes for which mutations occur significantly more frequently in primary tumors of patients with positive lymph node status than in primary tumor samples of patients with negative lymph node status. ** significance *p*-value < 0.05. * *p*-value < 0.1.

## Data Availability

Data were obtained from the Hartwig Medical Foundation at https://www.hartwigmedicalfoundation.nl/en/applying-for-data/ (accessed on 20 January 2020) with the permission of the Hartwig Medical Foundation. Data from Armenia et al. [1] are available at http://www.cbioportal.org/study?id=prad_p1000 (accessed on 11 September 2020). The data from Gundem et al. [13] are available at https://static-content.springer.com/esm/art%3A10.1038%2Fs41586-020-2581-5/MediaObjects/41586_2020_2581_MOESM1_ESM.xlsx (accessed on 18 February 2021), and Kumar et al. [14] are available at https://www.cbioportal.org/study/summary?cancer_study_id=prad_fhcrc (accessed on 23 March 2021). The results here include the use of data from The Metastatic Prostate Cancer Project (https://mpcproject.org/, accessed on 17 April 2021), a project of Count Me In (https://joincountmein.org/, accessed on 17 April 2021). GoNetic software is available at https://github.com/gmiclotte/gonetic (accessed on 17 April 2021) for non-commercial academic use.

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
