# Peer review of "Network-Based Analysis to Identify Drivers of Metastatic Prostate Cancer Using GoNetic"

_cancers, 2021, doi:10.3390/cancers13215291_

Round 1

Reviewer 1 Report

The manuscript by Louise at al develops a prediction module named GoNetic based on probabilistic pathfinding that uses a weighted prior network and mutational information to identify drivers of metastatic prostate cancer. It’s a novel network-based method that is able to detect genes with a lower mutational rate as members of recurrently mutated sets of genes connected on a prior interaction network, and they perform the network-based driver identification on the Hartwig Medical Foundation prostate cancer dataset (HMF cohort). When applied to the HMF cohort, the GoNetic successfully recovered known primary and metastatic drivers of prostate cancer that carry a strong frequency-based driver signal in the HMF cohort. The authors also own a specific algorithm to GoNetic to avoid bias towards genes that are altered frequently in a cohort, thus genes that are less frequently mutated can be also used to identify the cancer hallmarks effectively.

The OnCompare and SomInaClust are also performed to compare the number of mutations in a gene, or in sets of genes, between two different groups of samples, and identify mutational hotspots in the genes prioritized with GoNetic, respectively. Finally, according to the mutational enrichment analysis based on GoNetic, they find the genes, TP53, AR, RB1, CTNNB1, CSPG4 and APC exhibiting a strong mutational enrichment signal in metastatic samples and this in both the HMF and ARM metastatic cohorts. And mutations in genes like TP53, TAF1L, BRCA2 and MYH11 are verified in LN+ patients, which indicates a poor prognostic for patients.

Some suggestions/questions:

  • For introduction, the importance of genes with a lower mutational rate in identifying metastatic prostate cancer should be described.
  • For methods, the authors construct the GoNetic module based on the dataset of HMF cohort, and the other modules described in the manuscript may base on other datasets, weather the efficiency to predict cancer hallmarks is associated with the sample source. I mean why not combine multiple datasets?    

Reviewer 2 Report

General comments:

The article introduced the modification of prior work to identify cancer driver pathways in a program called GoNetic.  The program was then used on prostate cancer and further analysis of the results was conducted.  While the work was interesting and useful, the expectation of the content based on the abstract was different than the remaining parts of the article.  The abstract portrayed the article to be more about the GoNetic but the bulk of the methods and results focused on the prostate cancer analysis.  An update of either the abstract and simple summary to include more of the analysis or trimming of the main text should be done.

While the content of the manuscript was informative, the organization could be better.  For example, the methods section starts with the HMF data and then the methods section ends with additional datasets.  Placing them together would flow better. Additionally, the results section appears to have much of the discussion intermixed, while the discussion within the results section was good a simpler results section would be helpful with the expansion of the discussion section to maintain the information presented.  Introduction of prostate cancer and the datasets should be included into the introduction as well.  The analysis of the GoNetic identified genes should be grouped under one section, i.e. group OnCompare, GO term, etc to make it easier to identify what biological meaning could be gleaned from the analysis of the identified genes. Finally, many of the sentences are long and could be shortened/condensed to be clearer.

Other specific comments:

  • Maybe better to split Figure 1 into two figures, one for the methods section and then one for the results to have necessary visualization close to the main text.
  • Make Figure S1 and S2 into one table (not a figure) and place into main text
  • Create a venn diagram to highlight overlap the results of GoNetic with a few of the other methods (Gene lists from Figure S1 or FigureS2) Help show how consistent the results between the different methods are
  • It would be helpful to see Figure S7 panel A and Figure S8 in main text
  • There is a blank sheet in the middle of the SupplementaryTables file
  • Missing Subsection 3.3 (Goes from 3.2 -> 3.4)
  • Move Figure 2 to methods section not within the Results
  • HMF need to be referenced (line 78)
  • Reactome network (line 189) was not referenced

Round 2

Reviewer 2 Report

The concerns were adequately address.

Author Response

Not applicable, there are no unresolved comments from reviewer 2.